# Nutrition and Psoriasis

**DOI:** 10.3390/ijms21155405

**Published:** 2020-07-29

**Authors:** Naoko Kanda, Toshihiko Hoashi, Hidehisa Saeki

**Affiliations:** 1Department of Dermatology, Nippon Medical School, Chiba Hokusoh Hospital, Inzai, Chiba 270-1694, Japan; 2Department of Dermatology, Nippon Medical School, Bunkyo-Ku, Tokyo 113-8602, Japan; t-hoashi@nms.ac.jp (T.H.); h-saeki@nms.ac.jp (H.S.)

**Keywords:** psoriasis, nutrition, interleukin-17, vitamin D, *n*-3 polyunsaturated fatty acid, saturated fatty acid, short chain fatty acid, regulatory T cell, dysbiosis

## Abstract

Psoriasis is a chronic inflammatory skin disease characterized by accelerated tumor necrosis factor-α/interleukin-23/interleukin-17 axis, hyperproliferation and abnormal differentiation of epidermal keratinocytes. Psoriasis patients are frequently associated with obesity, diabetes, dyslipidemia, cardiovascular diseases, or inflammatory bowel diseases. Psoriasis patients often show unbalanced dietary habits such as higher intake of fat and lower intake of fish or dietary fibers, compared to controls. Such dietary habits might be related to the incidence and severity of psoriasis. Nutrition influences the development and progress of psoriasis and its comorbidities. Saturated fatty acids, simple sugars, red meat, or alcohol exacerbate psoriasis via the activation of nucleotide-binding domain, leucine-rich repeats containing family, pyrin domain-containing-3 inflammasome, tumor necrosis factor-α/interleukin-23/interleukin-17 pathway, reactive oxygen species, prostanoids/leukotrienes, gut dysbiosis or suppression of regulatory T cells, while *n*-3 polyunsaturated fatty acids, vitamin D, vitamin B12, short chain fatty acids, selenium, genistein, dietary fibers or probiotics ameliorate psoriasis via the suppression of inflammatory pathways above or induction of regulatory T cells. Psoriasis patients are associated with dysbiosis of gut microbiota and the deficiency of vitamin D or selenium. We herein present the update information regarding the stimulatory or regulatory effects of nutrients or food on psoriasis and the possible alleviation of psoriasis by nutritional strategies.

## 1. Introduction

Psoriasis is a chronic inflammatory skin disease characterized by an accelerated tumor necrosis factor-α (TNF-α)/interleukin-23 (IL-23)/IL-17 axis, and hyperproliferation and aberrant differentiation of epidermal keratinocytes [1]. The accelerated TNF-α/IL-23/IL-17 axis is the major pathomechanism of psoriasis [2] (Figure 1); dendritic cells (DCs) activated by various stimuli in the lesional skin secrete TNF-α which acts on themselves in an autocrine manner and induces IL-23 secretion. The IL-23 promotes the proliferation and survival of type 17 helper T (Th17) cells. The activated Th17 cells overproduce IL-17A or IL-22 which act on keratinocytes and induce their proliferation and production of TNF-α, antimicrobial peptides, or chemokines C-X-C motif ligand 1 (CXCL1)/8, C-C-motif ligand 20 (CCL20), which further recruit neutrophils, lymphocytes, or monocytes. The activation of keratinocytes by IL-17A or TNF-α induces the expression of keratins 6 and 16, which are associated with acanthosis and reduced turnover time in the epidermis [3]. Innate immune cells like type 3 innate lymphoid cells, γδT cells, or invariant natural killer T cells also produce IL-17A and are involved in the development of psoriasis. Moreover, 6% to 42% of psoriasis patients are associated with arthritis, called psoriatic arthritis (PSA) [1]. Psoriasis patients are frequently associated with obesity, diabetes mellitus, dyslipidemia, cardiovascular diseases, and inflammatory bowel diseases (IBDs). Evidence suggests that obesity is a risk factor for psoriasis, aggravates existing psoriasis, and that weight reduction may improve the severity of psoriasis in overweight patients [4]. Obesity is a chronic low level inflammatory state, and visceral fat-derived adipokines like TNF-α, leptin or visfatin induce the production of psoriasis-promoting antimicrobial peptides, human β-defensin-2/3 or chemokines, CXCL8/10, CCL20 in epidermal keratinocytes, and link metabolic syndromes to psoriasis [5,6,7]. A variety of genetic and environmental factors are involved in the pathogenesis of psoriasis and its comorbidities [8,9]. The environmental factors include dietary habits. The epidemiological studies revealed that psoriasis patients showed unbalanced dietary habits like higher intake of total fat, simple carbohydrates and lower intake of proteins, complex carbohydrates, monounsaturated fatty acid, *n*-3 polyunsaturated fatty acids (PUFAs), vegetables, and fibers, compared to healthy controls [10]. Psoriasis patients showed lower intake of Mediterranean dietary components (extra virgin olive oil, fruits, fish, and nuts) compared to healthy controls [11]. In vivo murine studies showed that the intake of a Western diet with high fat and high simple sugars aggravated the psoriasiform dermatitis induced by imiquimod (IMQ) [12]. It is suggested that certain nutrients or food exacerbate psoriasis such as saturated fatty acids (SFAs), simple sugars, red meat or alcohol while the others ameliorate psoriasis, like vitamin D, vitamin B12, *n*-3 PUFAs, dietary fibers, genistein, selenium, short chain fatty acids (SCFAs), or probiotics (Figure 2).

In this article, we review the recent studies regarding the stimulatory or regulatory effects of individual nutrients or food on psoriasis, and the possible alleviation of psoriatic symptoms by nutritional strategies.

## 2. The Nutrients or Food Related to Psoriasis

### 2.1. Fat

#### 2.1.1. SFAs

SFAs, such as palmitic acid or stearic acid, are rich in butter or red meat. The increased intake of SFAs is proposed to be a risk factor of obesity, dyslipidemia, or cardiovascular diseases [13]. The high-fat diet rich in SFAs exacerbates the IMQ-induced psoriasiform dermatitis in mice [13,14]; SFAs activate nucleotide-binding domain, leucine-rich repeats containing family, pyrin domain-containing-3 (NLRP3) inflammasome which generates active IL-1β and IL-18 in CD11c+ macrophages [14,15,16]. The increased IL-1β promotes the expression of CCL20 in the epidermis, leading to the accumulation of Th17 cells and γδT17 cells into the skin lesions [13,14,17]. SFAs induce the expression of epidermal-type fatty acid-binding protein (E-FABP), an intracellular chaperone coordinating lipid trafficking, and the E-FABP further couples lipid droplet formation and NLRP3-apoptosis-associated speck-like protein containing a caspase recruiting domain (ASC)-Caspase-1 inflammasome activation in CD11c+ macrophages [18]. SFAs also act on keratinocytes and endothelial cells, and induce their expression of CCL20, which may be related to the increased recruitment of Th17 cells and γδT17 cells into IMQ-induced psoriasiform dermatitis [13].

#### 2.1.2. *n*-6 PUFAs

The essential *n*-6 PUFA linoleic acid is rich in vegetable oils or margarines. Relations of linoleic acid consumption and psoriasis are still controversial. Linoleic acid undergoes metabolic conversion to arachidonic acid, which is incorporated in the cell membrane. When tissues are exposed to stimuli like injuries or cytokines, arachidonic acid is released from membranes and converted into lipid mediators by various enzymes. Prostanoids and leukotrienes (LTs) are the main lipid mediators derived from arachidonic acid, and might promote psoriasis [13]. Among prostanoids, thromboxane A_2_ (TXA_2_) binds TXA_2_ receptor (TP), and the level of TXA_2_ is increased in murine IMQ-induced psoriasiform dermatitis [19]. Stimulation of TP in γδT17 cells enhanced IL-23-induced production of IL-17A while inhibitors of TXA_2_ synthase ameliorated the IMQ-induced psoriasiform dermatitis and suppressed the IL-17 production by γδT17 cells [19]. TP-deficient mice showed the reduced inflammation with decreased number of γδT17 cells in the IMQ-induced dermatitis [19].

Another prostanoid, prostaglandin E_2_ (PGE_2_) also promotes the IMQ-induced psoriasiform dermatitis; PGE_2_ produced by fibroblasts promotes the production of IL-23 in dendritic cells (DCs) [20]. IL-23 induces the expression of cyclooxygenase-2 (COX-2) in Th17 cells, and induces the production of PGE_2_, which acts back on the PGE_2_ receptors, EP2 and EP4 in these cells and enhances IL-23-induced expression of IL-23R by activating signal transducer and activator of transcription 3 (STAT3), cyclic AMP-responsive element binding protein 1, and NF-κB through cyclic AMP-protein kinase A signaling [21]. This PGE_2_ signaling also induces the expression of various inflammatory Th17 signature genes like *IL17a, IL17f, IL18r1*. In IL-23-induced psoriasis model mice, the deletion of EP2 and EP4 in T cells inhibited the accumulation of Th17 cells and abolished the dermatitis [13,21].

Among LTs, LTB_4_ binds its receptor BLT1 on neutrophils, inducing their infiltration in the IMQ-induced psoriasiform dermatitis. The deletion of BLT1 or inhibition of LTB_4_ synthesis impaired the neutrophil infiltration and ameliorated the IMQ-induced dermatitis [13].

#### 2.1.3. *n*-3 PUFAs

*n*-3 PUFAs, eicosapentaenoic acid (EPA) and docosahexaenoic acid (DHA), rich in fish are metabolized into resolvin E1 and D1, respectively. Barrea et al. reported that psoriasis patients showed lower intake of *n*-3 PUFAs compared to healthy controls [10]. Serum *n*-3 PUFA level in psoriasis patients is inversely correlated with psoriasis area and severity index (PASI) [22]. *n*-3 PUFAs have been suspected to have anti-psoriatic effects. It is reported that *n*-3 PUFAs inhibit Th17 differentiation: DHA-treated DCs showed the decreased expression of costimulatory molecules, CD40, CD80, CD86, reduced IL-12, IL-23, IL-6 secretion, and exerted a reduced ability to induce Th1/Th17 differentiation [23]. DHA-rich diet ameliorated Th17-mediated inflammation in experimental allergic encephalitis model mice [23]. Resolvin E1 ameliorated the IMQ-induced psoriasiform dermatitis and reduced the expression of IL-23 and IL-17 in the lesions [24]. Resolvin E1 antagonized the LTB_4_-induced IL-23 production and migration of DCs, Th17 cells and γδT17 cells. These antagonistic effects of resolving E1 were mediated by binding to LTB_4_ receptor, BLT1. Resolvin D1 alleviated the IMQ-induced psoriasiform dermatitis and reduced the expression of IL-23, IL-17A, IL-17F, IL-22, and TNF-α in the lesions [25]. The effects of resolvin D1 were mediated by the inhibition of NF-κB, c-Jun N-terminal kinase, extracellular signal-regulated kinase and p38 mitogen-activated protein kinase (MAPK) and through lipoxin A4 receptor/formyl-peptide receptor 2 [25].

#### 2.1.4. SCFAs

SCFAs are the fermentation products of dietary fiber in the colon, and enter circulation and reach the organs outside of intestines, including the skin. SCFAs like butyrate, propionate and acetate, regulate inflammation in the intestines. Skin commensals like *Cutibacterium acnes* also produce SCFAs which might also exert a similar regulatory function in the skin [26]. SCFAs act via G-protein-coupled receptors (GPCRs) GPR109a, GPR41, or GPR43. SCFAs promote the activity of regulatory T cells (Tregs) and are suggested to ameliorate psoriasis [27]. SCFAs induce the production of transforming growth factor-β1 (TGF-β1) in intestinal epithelial cells [28], which can contribute to de novo induction of peripheral Tregs. SCFAs act on intestinal DCs via GPR109a and inhibit IL-23 expression and induce the expression of anti-inflammatory genes [29,30]. SCFAs, especially butyrate, induce the differentiation of thymic Tregs by promoting the expression of a transcription factor autoimmune regulator (*Aire*) in medullary thymic epithelial cells via GPR41 [31]. SCFAs also stimulate the differentiation of naïve CD4^+^ T cells into peripheral Tregs through histone H3 acetylation of the *Foxp3* gene intronic enhancer by inhibiting histone deacetylase [32]. Topical application of sodium butyrate in IMQ-induced psoriasiform dermatitis, reduced the inflammation, downregulated IL-17 expression, and induced *IL-10* and *Foxp3* transcripts [33]. Tregs isolated from the blood of psoriasis patients were reduced in their suppressive activity, which was normalized by sodium butyrate. The ex vivo analysis showed that sodium butyrate restored the reduced Treg number, IL-10 and Foxp3 expression as well as normalized the enhanced expression of IL-17 and IL-6 in human psoriatic skin lesions [33]. Lesional and non-lesional psoriatic skin showed the decreased expression of GPR109a and GPR43 on keratinocytes in comparison with control skin [34]. The reduced expression of both receptors was restored by topical application of sodium butyrate [34]. Sodium butyrate also enhanced terminal differentiation of keratinocytes and downregulated their proliferation via inhibiting histone deacetylase [35]: sodium butyrate in vitro increased the mRNA levels of filaggrin and transglutaminase A, and promoted cornified envelope formation in normal human keratinocytes [35].

### 2.2. Carbohydrates

#### 2.2.1. Simple Sugars

The excessive intake of simple sugars like sucrose may exacerbate psoriasis [36]. High-fructose-fed rats showed the increased serum IL-17F levels compared to control rats [37]. High glucose intake in mice exacerbated autoimmune colitis and experimental autoimmune encephalomyelitis [38]. High amounts of glucose specifically promoted Th17 differentiation by activating TGF-β from its latent form through upregulation of mitochondrial reactive oxygen species (ROS) in T cells [38].

Feeding with Western diet containing high levels of fat and of simple sugars in mice for 12 to 16 weeks aggravated the IMQ-induced psoriasiform dermatitis while high-fat, low-sugar diet did not [12]. The Western diet-fed mice showed the enhanced epidermal thickness, increased expression of neutrophil markers, *Cxcl2*, *Ly6g* mRNAs, Gr-1 protein and developed more Munro’s micro abscesses, and higher expression of NLRP3 and IL-1β in response to IMQ. The Western diet-fed mice showed the increased baseline expression of *IL-17a* without IMQ treatment. The mice fed with a high-fat, low-sugar diet showed more body weight gain, but less inflammation in response to IMQ than Western diet-fed mice, indicating that obesity is not sufficient and sugar content of the Western diet is a critical factor for the enhancement of dermatitis.

The short-time (4 weeks) feeding with Western diet promoted the accumulation of IL-17A-producing γδT cells with enriched IL-23 receptor expression in IMQ-induced psoriasiform dermatitis [39]. The short-term Western diet-promoted dermatitis was attenuated by supplementation with a bile acid (BA) sequestrant, cholestyramine or by antibiotic cocktail, indicating the involvement of BA pathway or of gut dysbiosis, respectively. Western diet intake causes gut dysbiosis and dysregulation in BA synthesis, either or both of which may mediate the induction of dermatitis. The gut microbiota and their metabolites can enter the circulation and be disseminated to the other organs including skin. Western diet in mice exacerbated the dextran sodium sulfate-induced colitis and induced the overgrowth of pro-inflammatory *E. coli*, decreased protective bacteria like Firmicutes, decreased SCFA production and its receptor GPR43 expression in the colon, and decreased forkhead box P3 (Foxp3)^+^ Tregs in mesenteric lymph nodes, leading to the enhanced systemic inflammation as well as colon [40]. Long-term (10 months) intake of Western diet in mice without IMQ induced dermatitis with enhanced Th17 and Th2 pathways, increased BA levels and enhanced expression of BA membrane receptors, Takeda G protein receptor 5 (TGR5) and sphingosine monophosphate receptor 2 (S1PR2) in the skin lesions [41]. BA sequestrant cholestyramine ameliorated the dermatitis. Activation of TGR5 induces pruritus via activation of transient receptor potential A1 in neurons [42], while the activation of S1PR2 in mice enhances vascular permeability and induces the expression of CCL2, tissue factor, vascular cell adhesion molecule-1, E-selectin in endothelial cells [43]. BAs produced from cholesterol in the liver are called primary BAs like cholic acid or chenodeoxycholic acid, which are then metabolized in the intestine by microbiota into secondary BAs like deoxycholic acid, lithocholic acid, or ursodeoxycholic acid [44]. BAs bind nuclear receptor farnesoid X receptor (FXR), primarily expressed in liver and intestine, and ubiquitously expressed membrane GPCRs, TGR5 and S1PR2. Gut microbiota changes the composition and amounts of BAs and modulates signaling via FXR, TGR5, and S1PR2. Conversely, BAs alter the composition of gut microbiota by promoting the growth of BA-metabolizing bacteria and inhibiting the growth of bile-sensitive bacteria [44]. Such crosstalk between BAs and gut microbiota may be related to the induction of dermatitis by Western diet.

The 2003–2006 National Health and Nutrition Examination Survey data in USA showed lower intake of simple carbohydrates in psoriasis patients compared to controls [45]. In contrast, Barrea et al. reported the higher intake of simple carbohydrates in male psoriasis patients compared to control males [10], indicating that gender, age or educational status may influence food choice behaviors.

#### 2.2.2. Complex Carbohydrates

Dietary fiber represents carbohydrates that are resistant to digestion in the small intestine and that undergo varying degrees of fermentation in the colon. Glucose-based resistant starch is abundant in grains or legumes, and highly fermentable. The dietary supplementation of fibers is reported to show systemic and intestinal anti-inflammatory effects [46]. The intake of fibers resulted in the improvement of IBDs and decreased levels of plasma inflammatory markers like C-reactive protein (CRP), IL-6, or TNF-α in parallel with the body weight loss [46]. Since a fiber-rich diet has lower energy density and obesity is a chronic low-grade inflammation status, the anti-inflammatory effects of dietary fiber may be partially via the body weight loss; however, obesity-unrelated mechanisms are also suggested. Dietary fibers, especially resistant starch are fermented in colon to generate SCFAs which may promote the activity of Tregs in the colon and also in the skin via circulation, leading to the regulation of inflammation in IBDs or psoriasis [47]. Fibers may also promote the growth of commensal bacteria and increase the resistance to the colonization of pathogenic bacteria, correcting the dysbiosis in gut microbiota [46]. Feeding with diet rich in fucoidan, seaweed fiber, in psoriasis model mice induced by *Traf3ip2* mutation, ameliorated symptoms of psoriasis-like dermatitis, scratching behaviors, and increased the secretion of mucin in ileum and of IgA in cecum, with alteration of the composition of gut microbiota [48].

### 2.3. Vitamins

#### 2.3.1. Vitamin D

There are two ways to meet vitamin D requirements in mammals: via dietary intake and via synthesis in skin by sun exposure. The food sources of vitamin D are cod liver oil, swordfish, salmon, tuna, sardines, beef liver, egg, or cheese. A whole-body exposure to UVB radiation inducing the minimal erythema dose for 15–20 min leads to the production of up to 10,000 IU of vitamin D while the recommended dietary allowance of vitamin D for adults ≤70 years is 600 IU/day [49]. The disease exacerbation of atopic dermatitis (AD) or psoriasis in winter may be at least partly due to the low sun exposure and subsequent low vitamin D production in the skin. The therapeutic effect of UVB therapy in the treatment of psoriasis may be, at least in part, mediated via UVB-caused synthesis of vitamin D in the skin; UVB therapy increased serum 25-hydroxyvitamin D level of psoriasis patients in parallel with disease improvement [50]. Vitamin D stimulates filaggrin synthesis, and AD patients are associated with *FLG* gene mutations. R501X and 2282del4 null mutations in *FLG* gene are associated with AD susceptibility among northern European populations with low sun exposure, but not among Greek and Egyptian cohorts with high sun exposure, indicating that *FLG* mutations are negatively selected in populations with high UV exposure and resultant high vitamin D synthesis in the skin [51].

It is reported that serum vitamin D levels are reduced in patients with psoriasis or PSA compared to controls [52,53]. In addition, the decreased expression of vitamin D receptor in psoriatic skin is correlated with the reduced expression of tight junction proteins like claudins, occludins, or Zonula Occludens-1 [52].

Vitamin D is a key modulator of inflammation. Vitamin D acts on monocytes/macrophages and down-regulates their production of TNF-α, IL-1β, IL-6, or IL-8 [52]. Vitamin D also dampens the differentiation, maturation, and antigen presentation of DCs. Vitamin D impairs the capacity of plasmacytoid DCs to induce T cell proliferation and interferon-γ (IFN-γ) secretion. Moreover, vitamin D inhibits the proliferation of T cells, reduces the number and IL-17A, IL-22 production in Th17 cells [54], while induces the generation of Tregs [55]. Vitamin D acts on keratinocytes and inhibits their proliferation and production of S100A7 and S100A15 which act as chemoattractants and alarmins. Topical vitamin D is effective for psoriasis [52]: it suppresses hyperproliferation in keratinocytes, and decreases the infiltration of Th17 cells and suppresses the expression of IL-12/23 p40, IL-1α, IL-1β, or TNF-α in the skin lesions. Topical vitamin D also normalizes the expression and topography pattern of integrins and other activation markers like intercellular adhesion molecule-1, CD26, and human leukocyte antigen-DR, which are altered in psoriatic skin lesions [52]. Vitamin D also upregulates the expression of late cornified envelope proteins (LC3A-E), which was reduced in psoriasis [52]. Several trials of oral vitamin D3 supplementation are under way for therapeutic use in psoriasis patients [56].

#### 2.3.2. Vitamin B12

Vitamin B12 is enriched in fish/shellfish (ark shell, oyster, clam, or salmon roe) or liver (beef, pork, or chicken). Vitamin B12 scavenges nitric oxides and ROS and thus protects various cells from inflammatory oxidative stress [57,58]. Vitamin B12 suppresses the phytohemagglutinin and concanavalin A-induced production of IL-6, IFN-γ, or IL-1β in human peripheral blood mononuclear cells [59]. ROS generation activates downstream NF-κB pathway, and Vitamin B12 suppresses the ROS-induced NF-κB activation and NF-κB-dependent production of inflammatory cytokines. Vitamin B12 suppressed the vincristine-induced activation of NADPH oxidase in rat spinal dorsal horn, and resultantly suppressed the phosphorylation of NF-κB p65 and TNF-α production and restored the reduced IL-10 production, in parallel with the reduction of atypical mitochondria in the sciatic nerve and restoration of the decreased intraepidermal nerve fibers induced by vincristine [60]. The effects of topical vitamin B12 on psoriasis are also reported [61]. However, a clinical trial of intramuscular injection of vitamin B12 together with conventional therapy did not reveal therapeutic efficacy for psoriasis [62].

#### 2.3.3. Vitamin A

The dietary sources of vitamin A (retinol) are livers, fish, eggs, or butter while provitamin A like β-carotene, which is converted to vitamin A after intestinal absorption, is enriched in green/yellow vegetables like carrots or spinach. Dietary vitamin A is absorbed in the intestine, delivered mainly to the liver, and to a lesser extent, to kidney, adipose tissues, or bone marrow [63]. Most of vitamin A actions depend on its active metabolites, retinoic acids (RAs) formed in the target tissues through intracellular oxidative metabolism [63]. Synthetic vitamin A derivatives, retinoids like etretinate or acitretin, after oral administration, are absorbed in small intestine, delivered to fat, liver, gut, or kidney, where they are metabolized to the active acid form RAs [64]. Retinoids are highly effective in the treatment of psoriasis. RAs act via retinoic acid receptors and retinoid X receptors. RAs act on keratinocytes in psoriasis lesions and normalize the hyperproliferation and stimulate the terminal differentiation [63]. RAs inhibit TNF-α production and reduce the mRNA levels of inducible nitric oxide synthase in keratinocytes [63]. RAs induce the generation of Tregs: RAs induce histone acetylation at *Foxp3* promoter and expression of Foxp3 protein in CD4^+^ T cells driven by mucosal DCs through costimulation via CD28 and dependently on TGF-β1 [65]. RAs enhance the TGF-β-dependent generation of Foxp3^+^ Tregs in peripheral CD4^+^ T cells by enhancing TGF-β signaling via increasing the expression and phosphorylation of Smad3, and simultaneously inhibits the development of Th17 cells by inhibiting the expression of IL-6Rα and IL-23R [66].

Treatment with retinoids might be associated with potential adverse effects, hyperostosis or tissue calcification [63]. It is reported that long-term treatment with retinoids, such as etretinate for 5 or 7 years [67,68], isotretinoin 4–6 years [69] resulted in enthesopathy with hyperossification in spines, ankles, pelvis, or knees, reminiscent of diffuse idiopathic skeletal hyperostosis (DISH) [67,70]. In DISH, certain factors including retinoids might stimulate mesenchymal stem cells at the periosteum or ligaments, and promote their abnormal osteoblast differentiation, leading to the hyperostosis at entheses [70]. The dietary intake of vitamin A in PSA patients was higher than that in psoriasis patients without arthritis in a Japanese study [71]. The relationship between higher vitamin A intake and the development or aggravation of PSA should further be investigated.

### 2.4. Genistein

Soybean is suggested to be a potential anti-psoriasis agent [72]. Isoflavones are phytoestrogens abundant in soybean, and genistein is the main isoflavone with potent anti-inflammatory activity. Topical genistein ameliorated the murine IMQ-induced psoriasiform dermatitis with the reduction of skin score, epidermal thickness, and of IL-1β, IL-6, TNF-α, CCL2, IL-17, and IL-23 expression in the skin lesions [73]. Genistein also in vitro suppressed the proliferation and expression of *IL-1**β*, *IL-6*, *IL-8*, *TNF-α*, *VEGFA (*gene for vascular endothelial growth factor A), *CCL2*, and *IL-23* mRNA in TNF-α-stimulated HaCaT cells [73]. Genistein suppressed STAT3 phosphorylation in IMQ-treated mice skin and TNF-α-stimulated HaCaT cells, and also inhibited Ικβ phosphorylation and nuclear translocation of NF-κB p65 in TNF-α-stimulated HaCaT cells [73].

Genistein also suppressed the ROS generation induced by TNF-α or lipopolysaccharide (LPS) in HaCaT cells by scavenging free radicals, enhancing the activity of antioxidant enzymes, and reducing the production of hydrogen peroxide [74]. It is thus supposed that genistein may attenuate ROS-mediated NF-κB activation and NF-κB-dependent inflammatory cytokine production in psoriatic lesional keratinocytes. Oral administration of genistein in psoriasis patients for 8 weeks impaired the transcription of genes overexpressed in psoriasis, *CXCL10*, *IL-6*, *STAT3*, *NFKB1*, *CCL4* while stimulated the transcription of gene repressed in psoriasis, *IL-1RN* in peripheral blood cells or lesional skin [75].

### 2.5. Selenium

Selenium is included broadly in fish/shellfish, eggs, poultry, or grains. Selenium is an essential trace element having anti-oxidative and immunoregulatory properties [76,77]. Selenium exerts various biological functions mainly as part of the selenium-containing proteins, selenoproteins, like glutathione peroxidases (GPxs) or thioredoxin reductases (TrxRs) [77]. GPxs are antioxidant enzymes that reduce ROS, such as hydrogen peroxide and lipid hydroperoxides. TrxRs catalyze the reduction of a wide range of substrates, including thioredoxin and protein disulfide isomerase. Selenium and selenoproteins regulate inflammation by altering eicosanoid production [77]. In dairy cattle with *E. coli*-induced mastitis, a selenium-sufficient diet decreased pro-inflammatory TXB_2_, PGE_2_, and LTB_4_ secretion in milk compared to cows with a selenium-deficient diet [78]. The overexpression of GPx4 in RBL-2H3 leukemia cells reduced the fatty acid hydroperoxides which activate 5-lipoxynase, and thus suppressed the activity of 5-lipoxygenase and subsequent production of LTB_4_ and LTC_4_ [79]. In macrophages or microglial cells, selenium supplementation suppressed LPS-induced NF-κB activation, NF-κB-dependent COX-2 expression, and PGE_2_ production, which was mediated via reducing ROS by selenoproteins GPxs or TrxRs [76,80,81]. The inhibition of MAPK pathways may also mediate the selenium-induced suppression of COX-2 expression [82]. Selenium supplementation in macrophages also enhanced the expression of hematopoietic PGD_2_ synthase and subsequent production of 15-deoxy-Delta12,14-prostaglandin J_2_, an anti-inflammatory eicosanoid which inhibits NF-κB, while suppressed the expression of NF-κB-dependent genes, microsomal PGE_2_ synthase and TXA_2_ synthase and subsequent production of PGE_2_ and TXA_2_ [83].

It is reported that serum selenium levels in psoriasis patients are lower than those in controls [84], indicating the imbalance of pro-/antioxidants in the patients.

### 2.6. Alcohol

Psoriasis may be triggered or worsened by alcohol intake [8,36,85]. A higher than average alcohol consumption is common among psoriasis patients [85]. A positive correlation between alcohol intake and the severity of psoriasis is reported for women [85]. Alcohol may initiate or aggravate the inflammation in psoriatic skin lesions [86]: ethanol increases the production of TNF-α in monocytes/macrophages, and increases lymphocyte proliferation and histamine release from mast cells [87]. Alcohol intake may induce pruritus by lowering its threshold. Alcohol-induced liver injury may impair skin barrier function via plasma TNF-α [88]: 5% or 10% ethanol in drinking water in mice for 8 weeks increased transepidermal water loss and reduced skin hydration, associated with decreased ceramide and type I collagen expression, increased tumor necrosis factor receptor 2 (TNFR2) expression in the skin, and increased plasma TNF-α level. The impairment of skin barrier was ameliorated by anti-TNF-α antibody. Ethanol enhanced the proliferation of HaCaT keratinocytes with enhanced expression of keratinocyte growth factor receptor, cyclin D1, and α5 integrin [89]. Ethanol enhanced uric acid-induced NLRP3 inflammasome activation and IL-1β production in human macrophages by reducing the expression of aryl hydrocarbon receptors and increasing that of thioredoxin-interacting proteins [90]. Chronic alcohol consumption in humans induces dysbiosis of gut microbiota, decreasing *Bacteroides* and increasing Proteobacteria, Fusobacteria and potentially pathogenic bacteria from *Prevotellaceae, Enterobacteriaceae, Vellionellaceae,* or *Streptococcaceae,* which may contribute to intestinal hyperpermeability and endotoxemia, and lead to systemic inflammation including the skin [91]. Chronic alcohol consumption in mice, 5% ethanol in drinking water for 10 weeks, exacerbated the IMQ-induced psoriasiform dermatitis with increased epidermal thickness and Th17-related cytokine expression [92]. The chronic ethanol consumption alone without IMQ-induced minimal epidermal infiltrates of T cells and the overexpression of IL-17A and CCL20. Low concentrations of ethanol in vitro induced the expression of CCL20 in murine epidermal keratinocytes. The results indicate that alcohol can promote Th17 pathway.

There is a link between increased alcohol intake and increased anxiety and depression [85]. Psoriasis patients have a 60% greater risk of dying due to alcohol-related causes, compared to controls, such as alcoholic liver diseases or psychological and behavioral disorders due to alcohol [85].

### 2.7. Red Meat

Red meat (beef, pork) abundantly contains SFAs, inducing NLRP3 inflammasome and IL-23/IL-17 pathway, and heme [93]. Dietary heme in mice induced gut dysbiosis and increased *Enterobacteriaceae* and *E. coli* while reduced Firmicutes and *Lactobacillus* in gut microbiota, and reduced the synthesis of butyrate which induces Tregs in colon, and exacerbated dextran sodium sulfate-induced colitis [94], which may also lead to the systemic inflammation including the skin. It is thus supposed that the excessive intake of red meat may aggravate the inflammation in psoriasis via the effects of SFAs and heme.

### 2.8. Probiotics

Probiotics are living microorganisms that confer health benefits to the host when administered in adequate amounts [95]. Most microorganisms as probiotics belong to the lactic acid-producing genera *Lactobacillus* and *Bifidobacterium* which can induce Tregs [95]. The administration of probiotics is supposed to give beneficial effects on psoriasis patients [95]. Certain gut microbes (*Bacteroides fragilis*, *Faecalibacterium prausnitzii*, *Clostridium* cluster VI and XIVa) and their metabolites (RAs, polysaccharide A, SCFAs) promote the activity and number of Tregs [27,95,96,97]. Some microbes and their metabolites enter circulation and can reach the skin, and coordinate epidermal differentiation, restore skin barrier, and balance the immune responses. Oral administration of *Lactobacillus brevis* SBC8803 in human subjects decreased transepidermal water loss and increased corneal hydration [98]. Oral administration of *Lactobacillus johnsonii* protects hairless mice against UV-induced contact hypersensitivity with reduction of epidermal Langerhans cells and increase of systemic IL-10 levels [99]. It is reported that patients with psoriasis and PSA show dysbiosis of gut microbiota [100]: the gut microbiota in these patients was less diverse compared to controls, and both groups showed the decrease in *Coprococcus* species. Gut microbiota in patients with psoriasis showed decrease in beneficial species *Parabacteroides* and *Coprobacillus*, which is common to IBD, while patients with PSA showed decrease in *Akkermansia* and *Ruminococcus*, protective taxa producing SCFAs. Another study reported that the gut of psoriasis patients showed less abundant *Faecalibacterium prausnitzii* producing butyrate and exerting anti-inflammatory action, compared to controls [101]. Decreased levels of bacteria of beneficial phyla may lead to poor regulation of intestinal and systemic inflammation including skin and joints.

Several studies show that supplementation with probiotics ameliorates skin inflammation in psoriasis model mice: *Lactobacillus pentosus* GMNL-77 administration reduced skin scores in IMQ-induced psoriasiform dermatitis and decreased the expression of TNF-α, IL-6, IL-23, IL-17A, IL-17F, and IL-22 in the skin lesions and decreased the numbers of Th17 and Th22 cells in the spleen [102]. The effects of GMNL-77 may be mediated via the suppression of intestinal antigen-presenting cells like CD103+ DCs and/or direct effects on T cells. *Bifidobacterium infantis* 35624 administration in psoriasis patients reduced plasma levels of TNF-α and CRP [103]. A randomized, double-blind, placebo-controlled trial was performed to determine the efficacy of probiotic mixture (*Bifidobacterium longum* CECT 7347, *B. lactis* CECT 8145 and *Lactobacillus rhamnosus* CECT 8361) for 90 psoriasis patients together with topical corticosteroid and calcipotriol [104]. At 12-week follow-up, 66.7% of patients in the probiotic group and 41.9% in the placebo group achieved Psoriasis Area and Severity Index (PASI) 75 score (*p* < 0.05). In the probiotic group but not in placebo group, there was a disappearance of *Rhodococcus*, a bacterium related to septicemia and biofilm production, and an increase in *Collinsella* and *Lactobacillus*, genera associated with better gut health, comparing initial and final time-points. The results supported that the probiotic blend alleviated psoriasis symptoms, and confirmed its efficacy in modulating the microbiota composition. Future studies using synbiotics, probiotics combined with dietary fibers, are promising.

The analysis of dietary habits in adult Japanese psoriasis patients revealed that they showed higher body mass indices (BMIs), higher intakes of fish/shellfish, pulses, vitamin B12, and vitamin D, and lower intake of meat, compared to those of healthy controls [71]. The results in Japanese patients are rather different from those in Western countries showing higher intake of fat and lower intake of fish though both share higher BMIs. The higher intake of vitamin D and vitamin B12 in Japanese patients may reflect the higher intake of fish/shellfish richly containing both vitamins. The results in Japanese psoriasis patients may reflect their taste preferring fish/shellfish and pulses to meat as side dishes. Alternatively, Japanese psoriasis patients may consciously avoid the dietary pattern to promote psoriasis and cardiometabolic diseases. To the contrary, Japanese psoriasis patients in the same study consumed more sugar/sweeteners compared to controls, and patients with high PASI showed higher intake of confections, compared to those with low PASI [71]. The dietary patterns in Japanese patients might thus be rather ambivalent: psoriasis- and cardiometabolic disease-attenuating patterns like lower meat intake and these favoring patterns like higher intake of sugar/sweeteners or confections. Whether such dietary patterns may be the ethnic characteristics of Japanese psoriasis patients should be identified by further extensive studies throughout Japan.

## 3. Possible Dietary Recommendation for Psoriatic Patients

Based on the stimulatory or regulatory effects of nutrients or food on psoriasis, it is proposed that dermatologists might evaluate the psoriatic patient′s current diet and nutritional status by consulting nutritionists, at the time of diagnosis. It is considered to guide the patients to the appropriate dietary plan if needed. The diets with appropriate calorie and balanced composition of nutrients might be proposed for the individual patients. Generally, psoriatic patients are proposed to take diet with appropriate composition of fat and sugar with sufficient intake of fish/shellfish, soybean, and dietary fibers, avoiding excess intake of red meat, simple sugars, and alcohol. The supplementation of *n*-3 PUFAs like DHA or EPA might be recommended. Patients with obesity might be considered to take low-calorie diet to reduce the weight. Psoriatic patients with low serum levels of vitamin D or selenium might be considered to take vitamin D or selenium supplementation, respectively. These dietary changes should be considered to complement the therapeutic effects of standard first line therapy for psoriasis or PSA, and also in parallel with the standard therapy for comorbidities like cardiometabolic diseases or IBDs.

## 4. Possible Alleviation of Psoriasis by Nutritional Strategies

Several trials of dietary changes for the alleviation of psoriasis have been done. Based on the review of nutritional strategies [105,106], dietary changes alone do not cause a large effect in psoriasis but may have some benefits supplementary to current first line treatments like biologics or systemic immunosuppressive medicine.

### 4.1. Low-Calorie Diet

Obesity is a chronic low-grade inflammation status, and is associated with the incidence and severity of psoriasis. Thus, low-calorie diet reducing weight may be helpful to improve psoriasis for the patients with obesity. A 24-week randomized controlled investigator-blinded clinical trial was conducted in 61 patients [107]. The efficacy of 2.5 mg/kg/day cyclosporine combined with low-calorie diet designed to achieve 5–10% weight loss (intervention group) was compared with cyclosporine alone (control group) in obese patients (BMI > 30) with moderate-to-severe psoriasis [107]. At week 24, the reduction in body weight was 7.0 ± 3.5% in intervention group vs 0.2 ± 0.9% in control group (*p* < 0.001), and PASI 75 was achieved in 66.7% in intervention group vs 29% in control group (*p* < 0.001). However, after discontinuation of cyclosporine, 80% of patients in intervention group returned to baseline PASI levels at week 52, despite the continued low-calorie diet, indicating that improvement was dependent on medication or medication adherence rather than diet. Several other controlled and uncontrolled clinical trials of low-calorie diet provided conflicting results [108,109,110,111,112], and are limited by small sample sizes. Future trials with larger sample sizes and with appropriate choice of concomitant treatment are needed to clarify the efficacy of low-calorie diet. A low-calorie diet may be recommended as an adjunct therapy in obese psoriasis patients but not in patients without obesity. The diet, not just with low-calorie but with balanced fat and sugar composition, should be administered.

### 4.2. Fish Oil or n-3 PUFA Supplements

Fish oil or *n*-3 PUFA dietary supplementation was theorized to aid in psoriasis patients. *n*-3 PUFA-supplemented diet improved PASI from 7.7 (baseline) to 5.3 at 3 months and 2.6 at 6 months compared to control (PASI: 8.9, 7.8, and 7.8, respectively, *p* < 0.05) [113]. However, the results of fish oil supplementation studies are overall conflicting in that some studies reported positive results [114,115,116,117] while others reported negative results [118,119,120]. Possibly the difference between intervention and control groups is small due to the high efficacy of concomitant treatment like biologics or narrowband UVB. Better-designed clinical trials with appropriate doses and sorts of fish oil are necessary to clarify the efficacy of this diet in psoriasis.

### 4.3. Vitamin D Supplementation

To explore the efficacy of oral vitamin D in psoriasis, an open-label study instructed 85 patients to ingest 0.5 µg of calcitriol daily followed by a 0.5 µg dose increase every two weeks for a total 6 months to three years [121]. Mean PASI improved from 18.4 (baseline) to 9.7 at 6 months and 7.0 at 36 months (*p* < 0.001). Other trials, however, report small sample sizes and inconsistent results [122,123,124]. Better designed studies with appropriate doses and types of vitamin D are required to evaluate the efficacy of vitamin D supplementation. The controlled trials limited to psoriasis patients with hypovitaminosis D might be beneficial.

### 4.4. Selenium Supplementation

Owing to low plasma levels of selenium in psoriasis patients, the efficacy of selenium supplementation was investigated by randomized, controlled trials and a clinical trial of selenium supplementation together with UVB therapy. These studies, however, showed no PASI improvements by selenium supplementation [125,126]. On the other hand, double-blind placebo-controlled clinical study on 28 PSA and 30 erythrodermic psoriasis patients showed the therapeutic effects of combined antioxidants, selenium, vitamin E, and coenzyme Q10 supplemented with conventional therapy [127]: PASI at day 30 in PSA patients was 16 ± 6 vs 29 ± 10 in intervention vs control group, respectively while that in erythroderma patients was 19 ± 4 vs 30 ± 5 (*p* < 0.05) [127]. Further better-designed studies with appropriate doses of selenium and possible combination with other antioxidants are required to evaluate the efficacy of selenium supplementation.

### 4.5. Gluten-Free Diet

Several studies documented the association of psoriasis and gluten sensitivity, celiac disease [128]. To investigate the effects of a gluten-free diet, an open label study for 30 psoriasis patients with elevated IgA antibodies to gliadin were started on a gluten-free diet together with continuation of topical or systemic psoriasis treatment for 3 months followed by a normal diet. After a 3-month gluten-free diet treatment, mean PASI score in all 30 patients improved from 5.5 ± 4.5 to 3.0 ± 3.6 (*p* < 0.001) [129]. After discontinuation of the gluten-free diet at 6 months, psoriasis was exacerbated in 18 out of 30 patients. Another study reported failure to improve after a 6-month gluten-free diet [130]. Due to the small sample size and lack of a comparison group, further controlled trials are necessary. Gluten-free diet may be recommended for psoriasis patients with anti-gliadin antibody.

## 5. Conclusions

We have reviewed the studies regarding the stimulatory or regulatory effects of nutrients or food on psoriasis and possible alleviation of psoriasis by nutritional strategies. SFAs, red meat, simple sugars, or alcohol exacerbate psoriasis and its comorbidities via the activation of NLRP3 inflammasome cascade or TNF-α/IL-23/IL-17 axis, generation of ROS, prostanoids/LTs, gut dysbiosis or suppression of Tregs. In contrast, *n*-3 PUFAs, vitamin D, vitamin B12, dietary fibers, SCFAs, genistein, selenium or probiotics ameliorate psoriasis or its comorbidities via the suppression of above inflammatory signaling pathways or the induction of Tregs. The manipulation of the disease-regulatory effects of nutrients or food may be useful for the management of psoriasis. Personalized diets could be proposed for individual patients based on their nutritional status and conditions of psoriasis and its comorbidities. Nutrition can be a key factor for the development and progress of psoriasis.

## Figures and Tables

**Figure 1 ijms-21-05405-f001:**
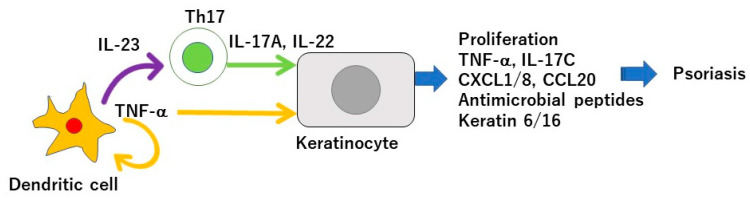
Tumor necrosis factor(TNF-α)/interleukin-23 (IL-23)/IL-17 axis in the pathogenesis of psoriasis. Dendritic cells activated by various stimuli in the lesional skin secrete TNF-α which acts on themselves in an autocrine manner and induces IL-23 secretion. The IL-23 promotes the proliferation and survival of Th17 cells. The activated Th17 cells overproduce IL-17A or IL-22 which act on keratinocytes and induce their proliferation and production of TNF-α, antimicrobial peptides, or chemokines C-X-C motif ligand 1 (CXCL1)/8, C-C-motif ligand 20 (CCL20), which further recruit neutrophils, lymphocytes, or monocytes. The activation of keratinocytes by IL-17A or TNF-α induces the expression of keratins 6 and 16, which are associated with acanthosis and reduced turnover time in the epidermis.

**Figure 2 ijms-21-05405-f002:**
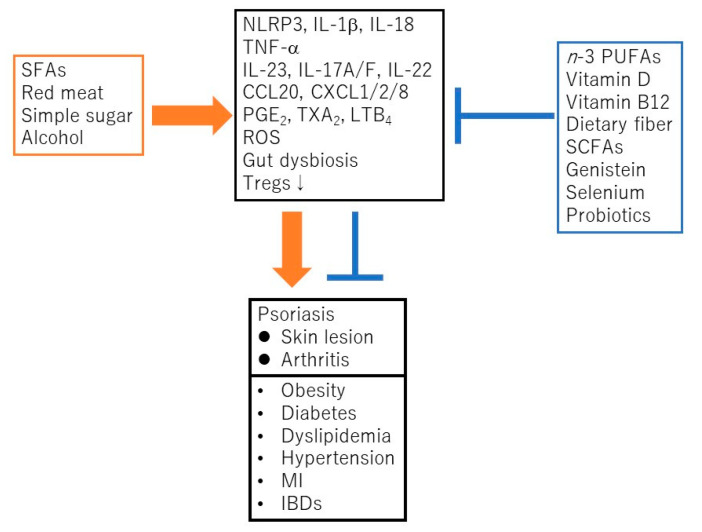
Saturated fatty acids (SFAs), red meat, simple sugars, or alcohol promote the development and progress of psoriasis and its comorbidities via the activation of nucleotide-binding domain, leucine-rich repeats containing family, pyrin domain-containing-3 (NLRP3) inflammasome cascade, TNF-α/IL-23/IL-17 axis, generation of reactive oxygen species (ROS), prostanoids/leukotrienes, gut dysbiosis, or suppression of regulatory T cells (Tregs). In contrast, *n*-3 polyunsaturated fatty acids (PUFAs), vitamin D, vitamin B12, dietary fibers, short chain fatty acids (SCFAs), genistein, selenium or probiotics ameliorate psoriasis or its comorbidities via the suppression of above inflammatory signaling pathways or the induction of Tregs. MI, myocardial infarction; IBD, inflammatory bowel disease; PGE_2_, prostaglandin E_2_; TXA_2_, thromboxane A_2_; LTB_4_, leukotriene B_4_.

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
