# Peer review of "Nutrition and Psoriasis"

_ijms, 2020, doi:10.3390/ijms21155405_

Round 1
Reviewer 1 Report
The review is very clear and well structured. I recommend making small changes in order to implement the final goal of the manuscript. Since the authors correctly explain that Psoriasis is a multifactorial disorder, in the section "2.3.1. Vitamin D", I recommend to insert data related to the role of vitamin D synthesized by exposure to sunlight. In fact, this topic is much discussed and applied to different dermatological diseases such as Atopic Eczema. In this regard, papers such as "Vitamin D and the Skin: An Update for Dermatologists; FLG (filaggrin) null mutations and sunlight exposure: Evidence of a correlation" could be cited. Finally, the conclusion could be improved by inserting an author's advice on the role of nutrition and the effects on Psoriasis. Could a personalized diet be proposed for each individual patient?
Author Response
Responses to comments by the reviewer
Reviewer 1
Comment: The review is very clear and well structured. I recommend making small changes in order to implement the final goal of the manuscript. Since the authors correctly explain that Psoriasis is a multifactorial disorder, in the section "2.3.1. Vitamin D", I recommend to insert data related to the role of vitamin D synthesized by exposure to sunlight. In fact, this topic is much discussed and applied to different dermatological diseases such as Atopic Eczema. In this regard, papers such as "Vitamin D and the Skin: An Update for Dermatologists; FLG (filaggrin) null mutations and sunlight exposure: Evidence of a correlation" could be cited.
Response: According to the reviewer’s suggestion, we have discussed the effects of vitamin D synthesized by sun exposure on skin diseases including psoriasis (section 2.3.1. marked). A whole-body exposure to UVB radiation inducing the minimal erythema dose for 15-20 minutes leads to the production of up to 10,000 IU of vitamin D while the recommended dietary allowance of vitamin D for adults ≦ 70 years is 600 IU/day. The disease exacerbation of atopic dermatitis (AD) or psoriasis in winter may be at least partly due to the low sun exposure and subsequent low vitamin D production in the skin. The therapeutic effect of UVB therapy in the treatment of psoriasis may be, at least in part, mediated via UVB-caused synthesis of vitamin D in the skin; UVB therapy increased serum 25-hydroxyvitamin D level of psoriasis patients in parallel with disease improvement. Vitamin D stimulates filaggrin synthesis, and AD patients are associated with FLG gene mutations. R501X and 2282del4 null mutations in FLG gene are associated with AD susceptibility among northern European populations with low sun exposure, but not among Greek and Egyptian cohorts with high sun exposure, indicating that FLG mutations are negatively selected in populations with high UV exposure and resultant high vitamin D synthesis in the skin. Papers regarding the effects of sun exposure-caused vitamin D are newly cited (refs 49-51).
Comment: Finally, the conclusion could be improved by inserting an author's advice on the role of nutrition and the effects on Psoriasis. Could a personalized diet be proposed for each individual patient?
Response: According to the reviewer’s suggestion, the conclusion is revised (section 5, marked). We have added the sentences below: Personalized diets could be proposed for individual patients based on their nutritional status and conditions of psoriasis and its comorbidities. Nutrition can be a key factor for the development and progress of psoriasis. The section of ‘Possible dietary recommendation for psoriatic patients’ (section 3), is newly added, and described the points to change the patients’ current diets possibly to alleviate the psoriasis symptoms (marked).
Reviewer 2 Report
I applaud the authors on conducting this highly relevant review article on nutrition and psoriasis. The authors have done a great job summarizing how diet can worsen or improve psoriasis which is highly clinically relevant to patients. I believe this review adds greatly to the field of psoriasis and I think this would be of great interest to the dermatology community. I would like to make a few comments to improve the manuscript.
- The authors mention type 17 helper T cell abnormal immune response in psoriasis. This would be a good place to also highlight the fact that TNF-a, IL 17 and IL 23 are commonly upregulated in psoriasis.
- At the end of the section on Vitamin A, the authors discuss treatment with retinoids can be associated with hyperostosis or tissue calcification. However, in current usage in dermatology, Acitretin is primarily used in a dose of 25 mg per day or less and no studies conclusively show this problem with skeletal toxicity with long term Acitretin use (utilizing the dose that is used currently). I would delete the phrase “indicating the possibility that vitamin A might promote hyperostosis”.
- I would change the title of section 3 “dietary intervention for psoriasis” as this seems like an over-promise. This current data is not strong enough to offer dietary changes as an “intervention” for psoriasis. Rather, the title should read something like “Possible mitigation of psoriatic symptoms through dietary changes” or something else along those lines.
- To make this more relevant to the dermatology provider, the authors may add a section summarizing their findings the implications for counseling patients in a clinical setting with a diagnosis of psoriasis regarding the possibility of changing their current diet. If not, an accompanying editorial may be worthwhile as well.
Author Response
Responses to the comments by the reviewer
Reviewer 2
Comment 1: The authors mention type 17 helper T cell abnormal immune response in psoriasis. This would be a good place to also highlight the fact that TNF-a, IL 17 and IL 23 are commonly upregulated in psoriasis.
Response: An accelerated TNF-α/IL-23/IL-17 axis is the major pathomechanism of psoriasis. The brief review of this axis is added to Introduction (section 1, marked) with relevant papers newly cited (refs 2, 3). Accordingly, figure showing the TNF-α/IL-23/IL-17 axis is newly added as Figure 1.
Comment 2: At the end of the section on Vitamin A, the authors discuss treatment with retinoids can be associated with hyperostosis or tissue calcification. However, in current usage in dermatology, Acitretin is primarily used in a dose of 25 mg per day or less and no studies conclusively show this problem with skeletal toxicity with long term Acitretin use (utilizing the dose that is used currently). I would delete the phrase “indicating the possibility that vitamin A might promote hyperostosis”.
Response: As suggested by the reviewer, the skeletal toxicity of long-term use of acitretin in current dose is not conclusive. We have deleted the phrase “indicating the possibility that vitamin A might promote hyperostosis.”, and just mention that the relationship between higher vitamin A intake and the development or aggravation of PSA should further be investigated (section 2.3.3., marked).
Comment 3: I would change the title of section 3 “dietary intervention for psoriasis” as this seems like an over-promise. This current data is not strong enough to offer dietary changes as an “intervention” for psoriasis. Rather, the title should read something like “Possible mitigation of psoriatic symptoms through dietary changes” or something else along those lines.
Response: As suggested by the reviewer, ‘the dietary intervention’ is overpromising. We have revised the title of this section as ‘Possible alleviation of psoriasis by nutritional strategies (section 4).’
Comment 4: To make this more relevant to the dermatology provider, the authors may add a section summarizing their findings the implications for counseling patients in a clinical setting with a diagnosis of psoriasis regarding the possibility of changing their current diet. If not, an accompanying editorial may be worthwhile as well.
Response: According to the reviewer’s suggestion, we have newly added the section ‘Possible dietary recommendation for psoriatic patients’ (section 3), and described the points to change the patients’ current diets possibly to alleviate the psoriasis symptoms. According to this revision, we have newly described the role of obesity in psoriasis in Introduction (section 1, marked), to clarify the significance of reducing weight in psoriasis patients with obesity, with relevant papers newly cited (refs 4-7).